# CUAD: An Expert-Annotated NLP Dataset for Legal Contract Review

**Dan Hendrycks**[*]
UC Berkeley

**Collin Burns**[*]
UC Berkeley

**Anya Chen**
The Nueva School

**Spencer Ball**
The Nueva School

## Abstract

Many specialized domains remain untouched by deep learning, as large labeled datasets require expensive expert annotators. We address this bottleneck within the legal domain by introducing the Contract Understanding Atticus Dataset (CUAD), a new dataset for legal contract review. CUAD was created with dozens of legal experts from The Atticus Project and consists of over 13,000 annotations. The task is to highlight salient portions of a contract that are important for a human to review. We find that Transformer models have nascent performance, but that this performance is strongly influenced by model design and training dataset size. Despite these promising results, there is still substantial room for improvement. As one of the only large, specialized NLP benchmarks annotated by experts, CUAD can serve as a challenging research benchmark for the broader NLP community.

## 1 Introduction

While large pretrained Transformers (Devlin et al., 2019; Brown et al., 2020) have recently surpassed humans on tasks such as SQuAD 2.0 (Rajpurkar et al., 2018) and SuperGLUE (Wang et al., 2019), many real-world document analysis tasks still do not make use of machine learning whatsoever. Whether these large models can transfer to highly specialized domains remains an open question. To resolve this question, large specialized datasets are necessary. However, machine learning models require thousands of annotations, which are costly. For specialized domains, datasets are even more expensive. Not only are thousands of annotations necessary, but annotators must be trained experts who are often short on time and command high prices. As a result, the community does not have a sense of when models can transfer to various specialized domains.

A highly valuable specialized task without a public large-scale dataset is contract review, which costs humans substantial time, money, and attention. Many law firms spend approximately 50% of their time reviewing contracts (CEB, 2017). Due to the specialized training necessary to understand and interpret contracts, the billing rates for lawyers at large law firms are typically around $500-$900 per hour in the US. As a result, many transactions cost companies hundreds of thousands of dollars just so that lawyers can verify that there are no problematic obligations or requirements included in the contracts. Contract review can be a source of drudgery and, in comparison to other legal tasks, is widely considered to be especially boring.

Contract review costs also affect consumers. Since contract review costs are so prohibitive, contract review is not often performed outside corporate transactions. Small companies and individuals consequently often sign contracts without even reading them, which can result in predatory behavior that harms consumers. Automating contract review by openly releasing high-quality data and fine-tuned models can increase access to legal support for small businesses and individuals, so that legal support is not exclusively available to wealthy companies.

---

[*]Equal contribution.

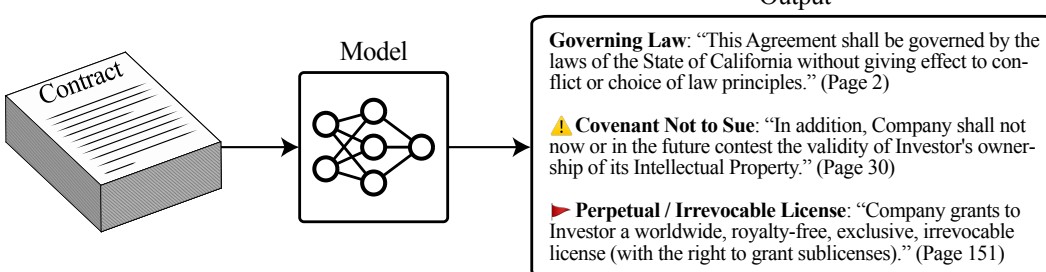

Figure 1: Contracts often contain a small number of important clauses that warrant review or analysis by lawyers. It is especially important to identify clauses that contain salient obligations or red flag clauses. It can be tedious and expensive for legal professionals to manually sift through long contracts to find these few key clauses, especially given that contracts can be dozens or even more than 100 pages long. The Contract Understanding Atticus Dataset (CUAD) consists of over 500 contracts, each carefully labeled by legal experts to identify 41 different types of important clauses, for a total of more than 13,000 annotations. With CUAD, models can learn to automatically extract and identify key clauses from contracts.

To reduce the disparate societal costs of contract review, and to study how well NLP models generalize to specialized domains, we introduce a new large-scale dataset for contract review. As part of The Atticus Project, a non-profit organization of legal experts, we introduce CUAD, the Contract Understanding Atticus Dataset (pronounced "kwad"). This dataset was created with a year-long effort pushed forward by dozens of law student annotators, lawyers, and machine learning researchers. The dataset includes more than 500 contracts and more than 13,000 expert annotations that span 41 label categories. For each of 41 different labels, models must learn to highlight the portions of a contract most salient to that label. This makes the task a matter of finding needles in a haystack.

CUAD is especially valuable because it was made possible with the collective effort of many annotators. Prior to labeling, law student annotators of CUAD attended training sessions to learn how to label each of the 41 categories, which included video instructions by and live workshops with experienced lawyers, detailed instructions, and quizzes. Before annotating contracts for our dataset, each law student annotator went through contract review training that lasted 70-100 hours. Annotators also adhered to over 100 pages of rules and annotation standards that we created for CUAD. Each annotation was verified by three additional annotators to ensure that the labels are consistent and correct. As a result of this effort, a conservative estimate of the pecuniary value of CUAD of is over $2 million (each of the 9283 pages were reviewed at least 4 times, each page requiring 5-10 minutes, assuming a rate of $500 per hour). This cost underscores the unique value of the CUAD dataset.

We experiment with several state-of-the-art Transformer (Vaswani et al., 2017) models on CUAD. We find that performance metrics such as Precision @ 80% Recall are improving quickly as models improve, such that a BERT model from 2018 attains 8.2% while a DeBERTa model from 2021 attains 44.0%. We also find that the amount of labeled training annotations greatly influences performance as well, highlighting the value of CUAD for legal contract review.

CUAD makes it possible to assess progress on legal contract review, while also providing an indicator for how well language models can learn highly specialized domains. CUAD is one of the only large, specialized NLP benchmarks annotated by experts. We hope these efforts will not only enable research on contract review, but will also facilitate more investigation of specialized domains by the NLP community more broadly. The CUAD dataset can be found at atticusprojectai.org/cuad and code can be found at github.com/TheAtticusProject/cuad/.

## 2   Related Work

### 2.1   Legal NLP

Researchers in NLP have investigated a number of tasks within legal NLP. These include legal judgement prediction, legal entity recognition, document classification, legal question answering, and legal summarization (Zhong et al., 2020). Xiao et al. (2015) introduce a large dataset for legal judgement prediction and Duan et al. (2019) introduce a dataset for judicial reading comprehension. However, both are in Chinese, limiting the applicability of these datasets to English speakers. Holzenberger et al. (2020) introduce a dataset for tax law entailment and question answering and

Chalkidis et al. (2019) introduce a large dataset of text classification for EU legislation. Kano et al. (2018) evaluate models on multiple tasks for statute law and case law, including information retrieval and entailment/question answering.

While legal NLP covers a wide range of tasks, there is little prior work on contract review, despite the fact that it is one of the most time-consuming and tedious tasks for lawyers. Chalkidis et al. (2017) introduce a dataset for extracting basic information from contracts and perform follow-up work with RNNs (Chalkidis et al., 2018). However, they focus on named entity recognition for a limited number of entities, a much simpler task than our own. The most related work to ours is that of Leivaditi et al. (2020), which also introduces a benchmark for contract review. However, it focuses exclusively on one type of contract (leases), it focuses on a smaller number of label categories, and it contains over an order of magnitude fewer annotations than CUAD.

## 2.2 NLP Models for Specialized Domains

Transformers have recently made large strides on natural language tasks that everyday humans can do. This raises the question of how well these models can do on *specialized* tasks, tasks for which humans require many hours of training. To the best of our knowledge, CUAD is one of only the large-scale NLP datasets that is explicitly curated for machine learning models by domain experts. This is also out of necessity, as there is no freely available source of contract review annotations that can be scraped, unlike for many other specialized domains.

There is some prior work applying machine learning to specialized domains. For example, machine translation has been a long-standing challenge that similarly requires domain expertise. However, unlike contract review, supervised data for machine translation is generally scraped from freely available data (Bojar et al., 2014). More recently, Hendrycks et al. (2021b) propose a challenging question answering benchmark that has multiple-choice questions from dozens of specialized areas including law, but the ability to answer multiple-choice legal questions does not help lawyers with their job. Similarly, there has been recent interest in applying language models to specialized domains such as math (Hendrycks et al., 2021c) and coding (Hendrycks et al., 2021a). Outside of NLP, in computer vision, machine learning has been applied to medical tasks such as cancer diagnosis that require specialized domain knowledge (Gadgil et al., 2021). These specialized tasks are not solved by current systems, which suggests the research forefront is in specialized domains.

## 3 CUAD: A Contract Review Dataset

**Contract Review.** Contract review is the process of thoroughly reading a contract to understand the rights and obligations of an individual or company signing it and assess the associated impact. Contract review is an application that is plausibly amenable to automation. It is widely viewed as one of the most repetitive and most tedious jobs that junior law firm associates must perform. It is also expensive and an inefficient use of a legal professional's skills.

There are different levels of work in contract review. The lowest level of work in reviewing a contract is to find "needles in a haystack." At this level, a lawyer's job is to manually review hundreds of pages of contracts to find the relevant clauses or obligations stipulated in a contract. They must identify whether relevant clauses exist, what they say if they do exist, and keep track of where they are described. They must determine whether the contract is a 3-year contract or a 1-year contract. They must determine the end date of a contract. They must determine whether a clause is, say, an anti-assignment clause or a most favored nation clause. We refer to this type of work as "contract analysis."

The highest level of work is to assess risk associated with the contract clauses and advise on solutions. At this level, a lawyer's business client relies on them to explain not only what each clause means, but also the implications such a clause has on its business and a transaction. This risk assessment work is highly contextual and depends on the industry, the business model, the risk tolerance and the priorities of a company. This is highly skilled work that is done by experienced in-house lawyers and law firm partners who are familiar with the clients' business. We refer to this type of work as "counseling."

To improve the lives of legal practitioners and individuals seeking legal assistance, our work aims to use machine learning models to automate the "contract review" work and the low level part of the "contract analysis" work.

| Category | Description |
|---|---|
| Effective Date | On what date is the contract is effective? |
| Renewal Term | What is the renewal term after the initial term expires? |
| Anti-Assignment | Is consent or notice required if the contract is assigned to a third party? |
| Governing Law | Which state/country's law governs the interpretation of the contract? |
| Perpetual License | Does the contract contain a license grant that is irrevocable or perpetual? |
| Non-Disparagement | Is there a requirement on a party not to disparage the counterparty? |

Table 1: A list of 5 of the 41 label categories that we cover in our dataset, along with short descriptions. Legal professionals deemed these labels to be most important when reviewing a contract. The Supplementary Materials contains the full list of categories.

**Labels.**   In designing our dataset for contract review, we consider clauses that would warrant lawyer review or analysis. We chose a list of 41 label categories that lawyers pay particular attention to when reviewing a contract. The labels are broadly divided into the following three categories:

- General information. This includes terms such as party names, document names, dates, governing laws, license grants, and renewal terms.

- "Restrictive covenants." These are considered some of the most troublesome clauses because they restrict the buyer's or the company's ability to operate the business.

- "Revenue risks." These include terms that may require a party to a contract to incur additional cost or take remedial measures.

We provide descriptions of sample label categories in Table 1 and include a full list in the Supplementary Materials.

**Task Definition.**   For each label category, we identify every clause in every contract that is most relevant to that label category. We then have models extract the relevant clauses from a contract by outputting the start and end tokens that identify the span of text that relates to that label category. Intuitively, models learn to highlight the portions of text that lawyers should attend to. We show example annotations in Figure 1.

**Dataset Statistics.**   CUAD contains 510 contracts and 13101 labeled clauses. In addition to belonging to 25 different types, contracts also have a widely varying lengths, ranging from a few pages to over one hundred pages. We show the distribution of contracts lengths in Figure 2. Most parts of a contract should not be highlighted. Labeled clauses make up about 10% of each contract on average. Since there are 41 label categories, this means that on average, only about 0.25% each contract is highlighted for each label.

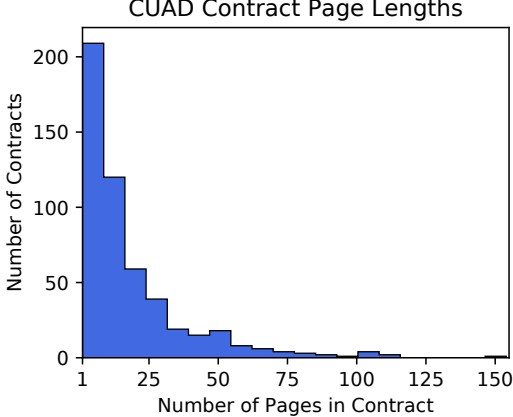

Figure 2: A histogram of the number of pages in CUAD contracts. Our dataset covers a diverse set of contracts. In addition to covering 25 different types of contracts, the contracts in our dataset also vary substantially in length, ranging from a few pages to well over one hundred pages.

**Supplementary Annotations.**   For each label category and each contract, we also include additional contract annotations that can be determined from the extracted clauses. For example, for the "Uncapped Liability" label category, we include the yes/no answer to the question "Is a party's liability uncapped upon the breach of its obligation in the contract?" for each contract, which can be answered from the extracted clauses (if any) for this label. To maintain consistency and simplicity, we do not focus on these supplementary annotations in this paper. We instead focus on evaluating the more challenging and time-consuming portion of this task, which is extracting the relevant clauses. However, we also release these additional annotations, which can further help apply models to contract review in practice.

**Contract Sources.**   Our dataset includes detailed annotations for 25 different types of contracts. We include a full list of contract types, along with the number of contracts of each type, in the Supplementary Materials.

We collected these contracts from the Electronic Data Gathering, Analysis, and Retrieval ("EDGAR") system, which is maintained by the U.S. Securities and Exchange Commission (SEC). Publicly traded and other reporting companies are required by the SEC rules to file certain types of contracts with the SEC through EDGAR. Access to EDGAR documents is free and open to the public. The EDGAR contracts are more complicated and heavily negotiated than the general population of all legal contracts. However, this also means that EDGAR contracts have the advantage of containing a large sample of clauses that are difficult to find in the general population of contracts. For example, one company may have only one or two contracts that contain exclusivity clauses, while EDGAR contracts may have hundreds of them.

**Labeling Process.** We had contracts labeled by law students and quality-checked by experienced lawyers. These law students first went through 70-100 hours of training for labeling that was designed by experienced lawyers, so as to ensure that labels are of high quality. In the process, we also wrote extensive documentation on precisely how to identify each label category in a contract, which goes into detail. This documentation takes up more than one hundred pages and ensures that labels are consistent.

## 4 Experiments

### 4.1 Setup

**Task Structure.** We formulate our primary task as predicting which substrings of a contract relate to each label category. Specifically, for each contract and label category, we have annotations for all of the substrings (if any) of that contract that should be highlighted. We then have a model learn the start and end token positions of the substring of each segment that should be highlighted, if any. This structure is similar to extractive question answering tasks such as SQuAD 2.0 (Rajpurkar et al., 2018) that allow for questions to have no answer. We consequently use the same model structure and training procedures as prior work on such tasks.

We finetune several pretrained language models using the HuggingFace Transformers library (Wolf et al., 2020) on CUAD. Because we structure the prediction task similarly to an extractive question answering tasks, we use the QuestionAnswering models in the Transformers library, which are suited for this task. Each "question" identifies the label category under consideration, along with a short (one or two sentence) description of that label category, and asks which parts of the context relate to that label category. To account for the long document lengths, we use a sliding window over each contract.

**Metrics.** Since most clauses are unlabeled, we have a large imbalance between relevant and irrelevant clauses. Therefore, we focus on measures that make use of precision and recall, as they are responsive to class imbalance.

Precision is the fraction of examples selected as important that are actually important, while

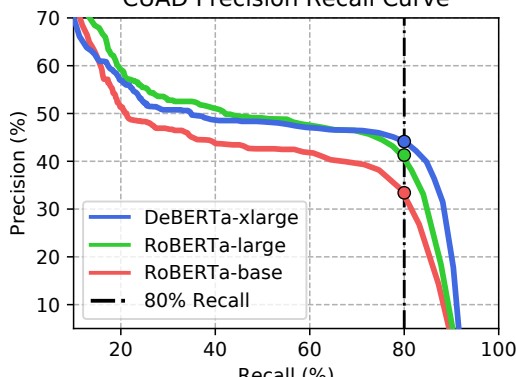

Figure 3: Precision-Recall curves for different models. We use the Area Under the Precision-Recall curve (AUPR) and Precision at 80% and 90% Recall as our primary metrics. There is a sharp dropoff in precision after around 80% recall, but this is improving with larger and more recent models such as DeBERTa-xlarge.

recall is the fraction of examples that are actually important that were selected as important. In our case, importance refers to a portion of a contract being relevant to a given label, which a human should review.

Precision and recall are defined in terms of true positives, false positives, and false negatives. A true positive is a ground truth segment of text that has a matching prediction. A false positive is a prediction that does not match with any ground truth segment. Finally, a false negative is when there is a ground truth segment of text that does not have a matching prediction.

| Model | AUPR | Precision @ 80% Recall | Precision @ 90% Recall |
|---|---|---|---|
| BERT-base | 32.4 | 8.2 | 0.0 |
| BERT-large | 32.3 | 7.6 | 0.0 |
| ALBERT-base | 35.3 | 11.1 | 0.0 |
| ALBERT-large | 34.9 | 20.9 | 0.0 |
| ALBERT-xlarge | 37.8 | 20.5 | 0.0 |
| ALBERT-xxlarge | 38.4 | 31.0 | 0.0 |
| RoBERTa-base | 42.6 | 31.1 | 0.0 |
| RoBERTa-base + Contracts Pretraining | 45.2 | 34.1 | 0.0 |
| RoBERTa-large | 48.2 | 38.1 | 0.0 |
| DeBERTa-xlarge | 47.8 | 44.0 | 17.8 |

Table 2: Results of NLP models on CUAD. We report the Area Under the Precision Recall curve (AUPR), Precision at 80% Recall, and Precision at 90% Recall. DeBERTa-xlarge has the best performance (44.0% Precision @ 80% Recall), which is substantially better than BERT-base (8.2% Precision @ 80% Recall), which highlights the utility in creating better models.

Each prediction comes with a confidence probability. With the confidences, we can smoothly vary the minimum confidence threshold we use for determining what to count as prediction (while always ignoring the empty prediction). We can then compute the best precision that can be achieved at the recall level attained at each confidence threshold. This yields a precision-recall curve, as shown in Figure 3. The area under this curve is then the Area Under the Precision Recall curve (*AUPR*), which summarizes model performance across different confidence thresholds.

We can also analyze model performance at a specific confidence threshold, giving rise to "Precision @ $X$% Recall" measures. As shown in Figure 3, if we threshold the confidence such that the model has 80% recall, then we can analyze the model precision at that threshold. Notice that as the recall increases, the precision decreases. Consequently Precision @ 90% Recall is less than Precision @ 80% Recall. Note having a precision of about 30% at this recall level means that a lawyer would need to read through about 2 irrelevant clauses for every 1 relevant clause selected as important by the model.

We determine whether a highlighted text span matches the ground truth with the Jaccard similarity coefficient. With the Jaccard similarity coefficient, we compute the overlap between the highlighted text and the ground truth. The Jaccard similarity coefficient is defined as $J(A, B) = \frac{|A \cap B|}{|A \cup B|}$, where $A$ is the set of words in an annotation, and $B$ is the set of words in an extracted prediction. To get the set of words in a string, we first remove punctuation and make the string lower case, then we separate the string by spaces. Note that $0 \le J(A, B) \le 1$, with $J(A, B) = 0$ when there is no intersection between $A$ and $B$, and $J(A, A) = 1$ for any non-empty set $A$. We use the threshold $0.5 \le J(A, B)$ for determining matches. We found that $0.5$ provides a qualitatively reasonable threshold, as it requires sufficiently high overlap for a span to be counted as a valid match.

**Models.** We evaluate the performance of BERT (Devlin et al., 2019), RoBERTa (Liu et al., 2019), ALBERT (Lan et al., 2020), and DeBERTa (He et al., 2020). BERT is a bidirectional Transformer that set state-of-the-art performance on many NLP tasks. RoBERTa improves upon BERT. RoBERTa uses the same architecture as BERT, but it was pretrained on an order of magnitude more data (160 GB rather than BERT's 16 GB pretraining corpus). ALBERT is similar to RoBERTa, but it uses parameter sharing to reduce its parameter count. DeBERTa improves upon RoBERTa by using a disentangled attention mechanism and by using a larger model size.

**Training.** More than 99% of the features generated from applying a sliding window to each contract do not contain any of the 41 relevant labels. If one trains normally on this data, models typically learn to always output the empty span, since this is usually the correct answer. To mitigate this imbalance, we downweight features that do not contain any relevant labels so that features are approximately balanced between having highlighted clauses and not having any highlighted clauses. For categories that have multiple annotations in the same document, we add a separate example for each annotation.

We chose a random split of the contracts into train and test sets. We have 80% of the contracts make up the train set and 20% make up the test set. In preliminary experiments we set aside a small validation set, with which we performed hyperparameter grid search. The learning rate was chosen from the set $\{3 \times 10^{-5}, 1 \times 10^{-4}, 3 \times 10^{-4}\}$ and the number of epochs chosen from the set $\{1, 4\}$.

In preliminary experiments we found that training for longer or using a learning rate outside this range degraded performance. We select the model with the highest AUPR found using grid search and report the performance of that model. For all experiments, we use the Adam optimizer (Kingma and Ba, 2015). Models are trained using 8 A100 GPUs.

## 4.2 Results

We show the results of fine-tuning each model in Table 2 and we show show precision-recall curves for three of these models in Figure 3. We find that DeBERTa-xlarge performs best, but that overall performance is nascent and has large room for improvment. DeBERTa attains an AUPR of $47.8\%$, a Precision at $80\%$ Recall of $44.0\%$, and a Precision at $90\%$ Recall of $17.8\%$. This shows that CUAD is a difficult benchmark. Nevertheless, these low numbers obscure how this performance may already be useful. In particular, recall is more important than precision since CUAD is about finding needles in haystacks. Moreover, $80\%$ recall may already be reasonable for some lawyers. The performance of DeBERTa may therefore already be enough to save a lawyer substantial time compared to reading an entire contract.

**Contracts Pretraining.** Since main driver of performance for language models is their large pretraining corpora, we determine whether domain-specific pretraining data can help with CUAD (Gururangan et al., 2020). We pretrain a RoBERTa-base model using the standard masked language modeling objective on approximately 8GB of unlabeled contracts collected from the EDGAR database of public contracts. As shown in Table 2, pretraining on several gigabytes of contracts increases AUPR by only about 3%. This shows that the high-quality annotated data in CUAD is currently far more valuable than orders of magnitude more unlabeled domain-specific data. Additionally, since the masked language modeling objective does not effectively leverage the large contract pretraining corpus, future algorithmic improvements in pretraining may be important for higher performance on CUAD.

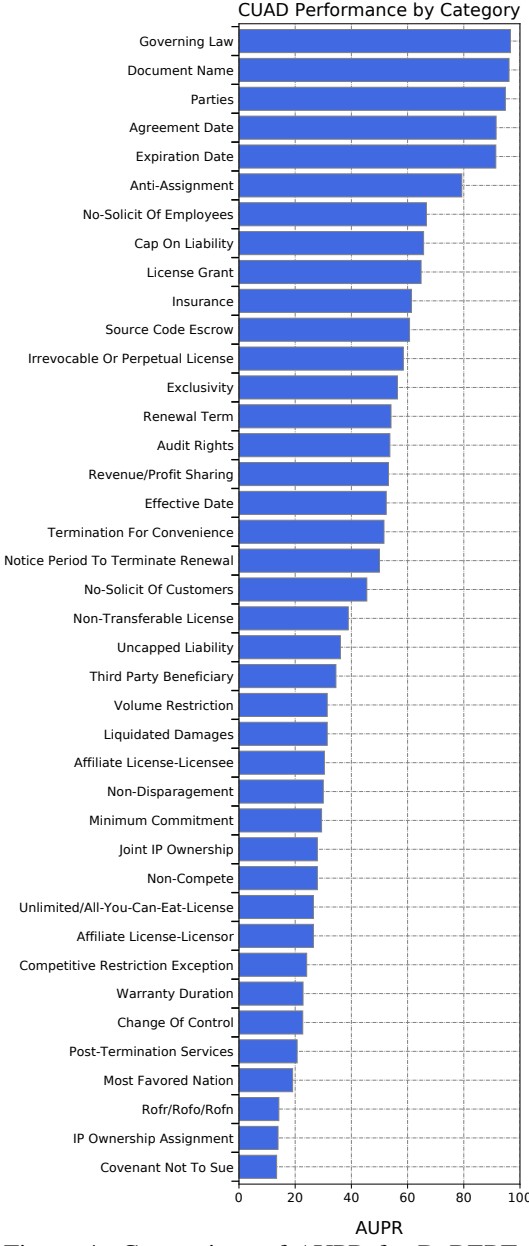

Figure 4: Comparison of AUPR for DeBERTa-xlarge across different label categories. While performance is high for some labels, it is has much room for improvement for other labels.

**Performance by Category.** In practice, models should be not only have strong overall performance, but also have strong performance in each individual label category. To compare performance across different categories, we compute the AUPR for DeBERTa-xlarge separately across all 41 categories, and show the results in Figure 4. We find that even though performance is high for some labels, it varies substantially by category, with some close to the ceiling of $100\%$ AUPR and others much lower at only around $20\%$ AUPR. This underscores that there is still substantial room for improvement.

**Performance as a Function of Model Size.** We now assess the effect of model size on performance. We measure the AUPR of various ALBERT models, ranging from ALBERT-base-v2 at 11 million parameters to ALBERT-xxlarge-v2 at 223 million parameters. Even though ALBERT-xxlarge-v2 has

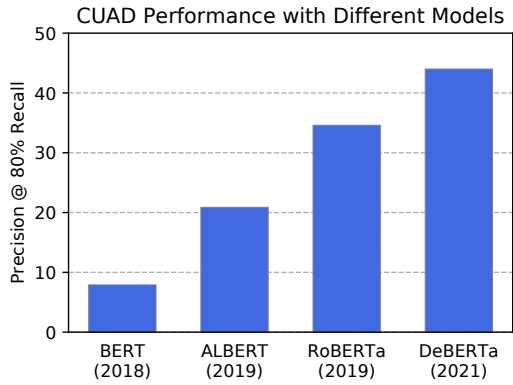

Figure 5: Performance on CUAD using chronologically aranged models. Each bar is an average of the performance of all models in each model class.

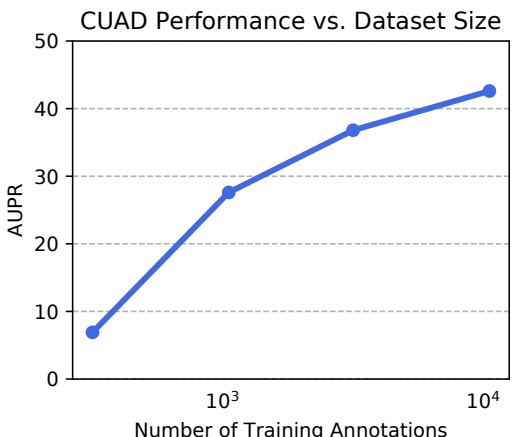

Figure 6: AUPR as a function of the number of training annotations for RoBERTa-base. This highlights the value of our dataset's size.

more than 20 times more parameters than its smallest version, it only performs around $3\%$ percent better. We find similar results with BERT as well; Table 2 shows only slight changes in the AUPR from BERT-base ($32.4\%$) to BERT-large ($32.3\%$).

On the other hand, model size seems to make an important difference in other cases. For example, RoBERTa-base ($42.6\%$) has noticeably lower performance than RoBERTa-large ($48.2\%$). There are also large differences in performance across different models, with DeBERTa performing far better than BERT. This suggests that while model size does not consistently help, model design can still be a path towards improving performance.

**Performance as a Function of Training Data.** We now assess how performance changes as a function of dataset size. We restrict our attention to RoBERTa-base and compute the AUPR as we vary the amount of training data. In particular, we test performance after training on $3\%$, $10\%$, $30\%$, and $100\%$ of the training contracts. To account for the smaller number of gradient updates that comes from having less data, we increase the number of training epochs in grid search to make the number of gradient updates approximately equal. For example, when we train on $30\%$ of the contracts, we consider grid search with the number of epochs in $\{3, 12\}$ instead of $\{1, 4\}$.

We show the results in Figure 6. We notice a substantial increase in performance as the amount of training data increases. For example, increasing the amount of data by an order of magnitude increases performance from $27.6\%$ to $42.6\%$, a $15\%$ absolute difference.

In fact, these gains in performance from just a single order of magnitude more data are comparable to the entire variation in performance across models. In particular, the best model (DeBERTa-xlarge) has an AUPR that is $15.4\%$ higher (in absolute terms) than that of the worst model in terms of AUPR. This indicates that data is a large bottleneck for contract review in this regime, highlighting the value of CUAD.

## 5 Conclusion

We introduced a high-quality dataset of annotated contracts to facilitate research on contract review and to better understand how well NLP models can perform in highly specialized domains. CUAD includes over $13,000$ annotations by legal experts across $41$ labels. We evaluated ten pretrained language models on CUAD and found that performance is promising and has large room for improvement. We found that data is a major bottleneck, as decreasing the amount of data by an order of magnitude cuts performance dramatically, highlighting the value of CUAD's large number of annotations. We also showed that performance is markedly influenced by model design, suggesting that algorithmic improvements from the NLP community will help solve this challenge. Overall, CUAD can accelerate research towards resolving a major real-world problem, while also serving as a benchmark for assessing NLP models on specialized domains more broadly.

**Acknowledgements**

A full list of contributors to the CUAD dataset is available at https://www.atticusprojectai.org/cuad. DH is supported by the NSF GRFP Fellowship. DH and CB are supported by Open Philanthropy Project AI Fellowships.

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
