# OpenReview forum: "CUAD: An Expert-Annotated NLP Dataset for Legal Contract Review"
_NeurIPS.cc/2021/Track/Datasets_and_Benchmarks/Round1 — NeurIPS 2021 Datasets and Benchmarks Track (Round 1)_

### Official Review · Reviewer_Nuhv · 2021-07-04
**Promising dataset on the highly specialized legal domain**

**Rating:** 6
**Confidence:** 3

**Strengths:**

1) This specialized domain of contract IR can be helpful to the legal community to build NLP systems. Moreover, from a research perspective, the resource would help QA models to test in the legal domain apart from the biomedical domain (as often done).

2) The experiments are fairly detailed for a dataset paper. They employ multiple transformer models and experiment with various setups that include domain-adaptive pre-training.


**Weaknesses:**

1) The labeling process, while detailed in the README, the statistics is not presented anywhere (to my knowledge). What were the agreements in the span finding? What were the common errors that human annotators made (it will be helpful in future model design too), and how were the disagreements resolved? Such details are important to be discussed in a dataset paper.

2) Benchmarked models are not multi-span extractors.

3) Error analyses and dataset exploration is not clearly presented.


**Additional Feedback:**

1) There should be some discussion on the discourse structure of these documents. Can models get better signals by knowing that the span of a particular label is present at point X in the contract, to predict another label?

2) Discussion on the variability of the content - for example, dates can be mentioned in a variety of ways and is dependent on the geographical location. How does this affect the span extraction process? Even if not from the model perspective, such features should be discussed (if present) from the data perspective.

3) Each label category can have multiple clauses in the contract. In the experiments, the models use a sliding window over the contract as context to extract clauses from. 1) Are these sliding windows released for fair benchmarking of future models? 2) [Line 255] “For categories that have multiple annotations in the same document, we add a separate example for each annotation.” - How many such examples were there? As the models are single-span extractors, they are basically incapable of predicting multiple spans in the contract. So, it would be good to mention how many multi-span clauses are typically there in a contract document.

4) Can there be a split with novel label categories in the test set? This could be a challenging few-shot or zero-shot scenario which might be useful for downstream purposes if lawyers want to look beyond the 41 categories.

5) Is the performance by category (Table 4) correlated to the class sizes? Or is there any specific pattern in the low-performing classes? Such error analysis would be helpful for a reader to better understand the dataset and the task.


**Clarity:**

Yeah, the paper seems to be well written and clear to understand.

Minor typo: [Line 219] a false negative is a when

**Correctness:**


Overall, there doesn't seem to be any major issues in correctness with the task framing.


**Documentation:**


The github repository, which releases the checkpoints, is the entry point for people to start working on the dataset. As such, the README can be made more verbose with better examples, model training and evaluation setup details, how to reproduce results and how to access the data.


**Ethics:**


There should be a clear discussion on the demographic assumptions of the dataset. As this is a highly specialized domain, there is bound to be a difference in the structure of contracts across different industries and geographical locations. These differences could also be in the language and the specialized jargon. As such, there should be a discussion on what demographics this dataset is based on. Also, are there any plans to sample a more diverse set in V2?


**Relation To Prior Work:**

Personally, I am not informed of the literature of legal NLP. However, the draft seems to cover a good amount of existing resources. One way to improve this discussion is to show some summarized comparisons between the dataset provided by Leivaditi et al. versus CUAD.

Section 2 could also cover, in addition to specialized resources, what are the popular techniques to adapt Transformer models towards these specialized tasks -- also from the lens of sample efficient training. And how do the experiments connect to these techniques?


**Summary And Contributions:**

This paper proposes a novel dataset for contract review. Contracts are legal documents that require human reading over some sections which are generally not delineated to the laymen. As such, the resource aims to annotate such salient sections of a contract (clauses) with respect to a curated list of classes. In other words, the task is that of information retrieval of text from a highly specialized domain - legal documents.

The paper also presents some experiments to benchmark the dataset. Over multiple transformer models, they perform question-answering style span extraction.

---

### Official Review · Reviewer_icsE · 2021-07-04
**Dataset made with an explicit motivation and substantial effort, accompanying high amount of budget**

**Rating:** 6
**Confidence:** 3
**Clarity:** The paper clearly describes the proce…

**Strengths:**

- This benchmark specializes in legal contract review, a domain of legal NLP that deals with consuming work
- The dataset is annotated on publicly available documents so as to guarantee the open usage and further modification and remix
- The benchmark is validated with up-to-date PLMs to check that the dataset is challenging enough as a benchmark, and it is also discovered that the performance is not merely boosted by conducting a pretraining with legal corpora, again proving the quality of the annotation

**Weaknesses:**

- It is sure that the dataset can make up a challenging benchmark, but it is not proven that the systems trained with the proposed dataset can truly alleviate the difficulty of lawyers reviewing the contracts, by an autonomous extraction of phrases. This makes opaque the motivation or purpose of constructing this benchmark. If most PLMs score 0.0 for precision@90% recall, then will the systems trained with the proposed dataset be supportive for the lives of lawyers? Though this is more related to human-computer interaction and may need some user study, I think the benchmark being challenging, ironically deters the utility of the proposed dataset.

**Additional Feedback:**

Some feedbacks are in the above. My recommendation is:
- ensuring that the trained systems are truly helpful to contract reviewing
- stating the ethical considerations explicitly, especially regarding the annotation bias of legal domain

**Correctness:**

I think the chosen evaluation scheme does make the benchmark challenging, but not sure if it is appropriate for checking the contract reviewing ability of the model. Maybe a human evaluation for a sampled, small size of dataset would also fit for checking the performance.

**Documentation:**

It maybe valuable to the community if the annotation guideline (at least in a short form) or the considerations in the tagging procedure are revealed. Other documentations are well organized in the paper and the web pages.

**Ethics:**

Though the construction process may not be reproducible due to the amount of budgets, revealing and distributing the details of dataset may compensate for such exclusivity of legal domain. One concern is that the model trained based on the corpus may incorporate the decisions of the law experts, which can be controversial for some times. I think some kind of statement is necessary as an ethical consideration part, that the annotations contain less controversial topics, or the annotations are cross checked to reduce the bias.

**Relation To Prior Work:**

Legal NLP and NLP models for specialized domains are appropriate prior work. However, it may be better if there is an explicit comparison between the prior work that deals with contract review, e.g., in form of table.

**Summary And Contributions:**

This paper demonstrates how the authors constructed an expert-annotated dataset regarding legal contract review. It is quite valuable material given that tremendous effort and expert knowledge would have been put in the construction phase, and its validity is partially checked by evaluation with up-to-date pretrained language models. The dataset is publicly open and will be a substantial contribution to the legal NLP community.

---

### Official Review · Reviewer_VsBr · 2021-07-06
**A dataset for legal contract review**

**Rating:** 7
**Confidence:** 4
**Correctness:** The data collection process and the e…

**Strengths:**

•	Good explanation of the need for a legal contract review dataset
•	All questions are human-authored, both with and without looking at the table. This makes the task very realistic.
•	The idea is quite novel, which will address many of the ambiguities faced with legal contracts

**Weaknesses:**

•	dataset's rather small.
•	what metric did you use to compare your work to others?
•	the literature is too narrow.

**Additional Feedback:**

Please use the highlighted in the weaknesses section to address a number of concerns in the paper.

**Clarity:**

The paper is very well written and understandable. The arguments for the need of the new dataset are presented eloquently.

**Documentation:**

The data collection is well documented.

**Ethics:**

No ethical issues

**Relation To Prior Work:**

The related works section explains previous work well but to a limited extent. It will be nice to compare your work with others (a table could help here).

**Summary And Contributions:**

The paper presents a new dataset for annotated contract to facilitate work related to contract review. The dataset showed interesting features, which could alienate several clauses that are usually common with most contracts. This is a well-written article and an interesting piece.

---

### Official Review · Reviewer_MEMP · 2021-07-07
**A specialized dataset in legal contract analysis with extensive comparisons of modern pre-trained LM models**

**Rating:** 8
**Confidence:** 4

**Strengths:**

- Legal contract reading is a a tedious and time-consuming job for junior lawyers, and the assistance with NLP models can greatly improve the efficiency of this job. Thus, the introduced benchmark dataset is a great resource for NLP researchers to work on the legal domain.

- The annotation of legal contract analysis requires expertise knowledge and is very costly. The benchmark dataset in this paper is annotated by volunteer legal experts and thus is precious to the NLP community from this perspective.

- The suite of experiments on evaluating the ability of pre-trained language models is comprehensive and the discussions of evaluation metrics is also clear. Experiments show that state-of-the-arts LM models still struggle in performing well on the legal domain, which involves fine-grained retrieval in a very long document. This further corroborates the value of this dataset in the NLP research.

**Weaknesses:**

- It's not very clear how the authors use sliding windows to process the whole document, which should be clarified. And does the retrieval of spans involve long-term dependency to previous tokens or do you process the document sentence by sentence or segments by segments?

- The processing of legal documents should be very related to long sequence modeling in the literature of efficient Transformer models, which should be discussed.

- A minor place is that when evaluating the model on different sizes of training data, authors try to match the update steps for different sizes, but for small size of data, this could cause overfitting.

**Additional Feedback:**

modified: 19 Jul 2021

**Clarity:**

The paper is very well written and structures with small typos in L266 and L394 in page 7.

**Correctness:**

The description of the construction of the dataset looks correct to me, but I am not sure if the 41 types of labels chosen for the dataset covers most of cases in general legal contract review.

**Documentation:**

Unfortunately, this seems to be lacking in some parts as far as I could see including the documentation and annotation details to support reproducibility, but authors mentioned they created a document of more than 100 pages.

**Ethics:**

N/A.

**Relation To Prior Work:**

The literature review in legal domain datasets as well as specialized datasets is clear and sufficient.

**Summary And Contributions:**

This paper introduces a domain specialized dataset for legal contract analysis, which is the first English dataset in this area and fills one blank of NLP datasets. The dataset is created by legal experts and with precise annotation documentation.
Extensive results with modern pretrained language models reveal the limitations of their performance in this specialized domain.

---

### Author Response · Authors · 2021-07-15
**Reply**

We thank the reviewers for their careful analysis of our paper, valuable feedback, and for their unanimously positive reviews.

> “It's not very clear how the authors use sliding windows to process the whole document, which should be clarified.”

To clarify, we use a context size of 384 and a stride of 128 to break up long documents into chunks, and process these chunks separately, as is shown in the training code that we provided in the GitHub repo.

> “the dataset is rather small”

While this dataset does not have on the order of 100,000 examples, more than 13,000 annotations is still large enough for effective finetuning and evaluation of language models. Many widely used NLP benchmarks are not as large, as 5 out of 9 of the GLUE datasets are not as large as CUAD. Moreover, collecting as many annotations as this in the legal domain, where expert annotators are expensive, was a very costly undertaking.

> “It is sure that the dataset can make up a challenging benchmark, but it is not proven that the systems trained with the proposed dataset can truly alleviate the difficulty of lawyers reviewing the contracts, by an autonomous extraction of phrases.”

Yes, models need to perform better before they can be relied on by attorneys. This is why we propose the task, as state-of-the-art models do not yet have strong performance, making it important for further progress to be made by the machine learning community. If you are instead concerned that even future models with strong performance will not be useful for lawyers, note that CUAD itself was initiated and designed by practicing attorneys. See the Attorney Advisors here: https://www.atticusprojectai.org/cuad It is consequently focused on a very particular task that attorneys agree can and should be automated.

> “The labeling process, while detailed in the README, the statistics is not presented anywhere (to my knowledge).”

To clarify, in addition to the details about the labeling process that we include in the main body (such as how each annotation was checked 4 times), we also provide many additional details about the labeling process in the supplementary materials.

> “Benchmarked models are not multi-span extractors.”

There are undoubtedly many model improvements that could be (and we hope will be!) made to improve performance. We deliberately chose to focus on baseline models customized for our task, but others can easily build on our work by more complicated models.

---

### Comment · Reviewer_VKwr · 2021-08-06
**An official ethics review**

This is a comment from an official ethics reviewer.

Two of the reviewers point out that there may be biases in the contracts themselves and in the attorney labelers. This is a valid concern, but true of any dataset and any domain. I do not believe there to be any extra concern in this dataset. My only recommendation is that the authors create a factsheet and release it with the dataset for additional transparency. This document can include a brief description of the demographic and other characteristics of the expert attorney labelers and how the pool of contracts was sampled. The acceptance of the paper should not be held up by ethical considerations.

---

### Comment · Program_Chairs · 2021-08-15
**Official Ethics review**

I agree that very robust meta-data (likely manifest as a factsheet) is necessary for this paper. There are a few other broader impacts considerations that the authors need to address in their final draft:

* This paper should contain a discussion of potential negative impacts. The introduction has all the traditional problems of a classic NeurIPS introduction in that it only highlights the positive societal impacts of the contribution and completely ignores the potential negative impacts. There are many trivial ways such a dataset could result in societal harm as I have enumerated below. I would like to see the authors apply their domain expertise to identifying risks that can't be identified by someone like myself without legal training.

* Some examples of obvious societal risks include:

* Such a dataset could help in the creation of clauses that outwit human and automated contract review.
* Such a dataset could disrupt existing career paths in law, resulting in more demographic inequality in top firms.
* The authors likely have used the labor of those who are going in debt to create a dataset that will reduce opportunities for them to get out of debt via entry-level legal jobs. How are the benefits of this dataset going to flow to the dataset creators rather than just the authors?
* Such a dataset works towards automating the task of contract review, but not the accountability of contract review services. One can easily imagine the low-cost services provided to small businesses etc. offering no accountability, thereby passing all the risk of classification errors onto the the client. In other words, the automation here would create more harm by encouraging people to trust a flawed system. This precise issue has resulted in a good portions of the harms created by computing technology, e.g. the elimination of privacy as we once knew it.

* The authors may also want to consider releasing a dataset with a license similar to those available at licenses.ai to attempt to control the use of the dataset.
* More generally, the authors should think more deeply about how the release of such a dataset advances the noble goals of democratizing the benefits of the legal system, and how it might actually put them at risk.

These issues need to be made transparent in the paper, but as long as they are transparent, I do not believe that the paper contains the exceptional ethical risks that would be necessary to withhold publication.

---

> ### Author Response · Authors · 2021-08-15
> **Datasheet is available**
>
> We have prepared a datasheet, though in hindsight we should have also included it in the supplementary material so that it is easier for reviewers to find, rather than just linking to it on the site.
>
> It is linked to here:
> https://www.atticusprojectai.org/cuad
>
> The datasheet and the README can also be found here:
> https://drive.google.com/drive/u/1/folders/1Yu-JnZj1LbVBfTdPiHfMDnaKZj4eqks8

---

### Decision · Program_Chairs · 2021-07-26

Accept